# Eight Weeks of Intermittent Exercise in Hypoxia, with or without a Low-Carbohydrate Diet, Improves Bone Mass and Functional and Physiological Capacity in Older Adults with Type 2 Diabetes

**DOI:** 10.3390/nu16111624

**Published:** 2024-05-26

**Authors:** Raquel Kindlovits, Ana Catarina Sousa, João Luís Viana, Jaime Milheiro, Bruno M. P. M. Oliveira, Franklim Marques, Alejandro Santos, Vitor Hugo Teixeira

**Affiliations:** 1Faculty of Nutrition and Food Sciences, University of Porto, FCNAUP, 4150-180 Porto, Portugal; nutri@raquelkin.com (R.K.); bmpmo@fcna.up.pt (B.M.P.M.O.); alejandrosantos@fcna.up.pt (A.S.); 2Research Center in Sports Sciences, Health Sciences and Human Development, CIDESD, University of Maia, 4475-690 Maia, Portugal; acsousa@umaia.pt (A.C.S.); jlviana@umaia.pt (J.L.V.); 3CMEP, Exercise Medical Centre Laboratory, 4150-044 Porto, Portugal; jaimemilheiro@yahoo.com; 4Centre of Research, Education, Innovation and Intervention in Sport, CIFI2D, Faculty of Sport, University of Porto, 4200-540 Porto, Portugal; 5Laboratory of Artificial Intelligence and Decision Support, Institute for Systems and Computer Engineering, Technology and Science (LIAAD, INESC-TEC), 4200-465 Porto, Portugal; 6Laboratory of Biochemistry, Department of Biological Sciences, UCIBIO, REQUIMTE, Faculty of Pharmacy, University of Porto, 4050-313 Porto, Portugal; franklim@ff.up.pt; 7Institute for Research and Innovation in Health, i3S, 4200-135 Porto, Portugal; 8Research Center in Physical Activity, Health and Leisure, CIAFEL, Faculty of Sports, University of Porto, FADEUP, 4200-540 Porto, Portugal; 9Laboratory for Integrative and Translational Research in Population Health, ITR, 4050-600 Porto, Portugal

**Keywords:** body composition, normobaric hypoxia, carbohydrates, diabetes, elderly, performance, bone

## Abstract

In an increasingly aging and overweight population, osteoporosis and type 2 diabetes (T2DM) are major public health concerns. T2DM patients experience prejudicial effects on their bone health, affecting their physical capacity. Exercise in hypoxia (EH) and a low-carbohydrate diet (LCD) have been suggested for therapeutic benefits in T2DM, improving bone mineral content (BMC) and glycemic control. This study investigated the effects of EH combined with an LCD on body composition and functional and physiologic capacity in T2DM patients. Older T2DM patients (*n* = 42) were randomly assigned to the following groups: (1) control group: control diet + exercise in normoxia; (2) EH group: control diet + EH; (3) intervention group: LCD + EH. Cardiopulmonary tests (BRUCE protocol), body composition (DEXA), and functional capacity (6MWT, handgrip strength) were evaluated. Body mass index (kg/m^2^) and body fat (%) decreased in all groups (*p* < 0.001). BMC (kg) increased in all groups (*p* < 0.001) and was significantly higher in the EH and EH + LCD groups (*p* < 0.001). VO_2peak_ improved in all groups (*p* < 0.001), but more so in the hypoxia groups (*p* = 0.019). Functional capacity was increased in all groups (*p* < 0.001), but more so in the EH group in 6MWT (*p* = 0.030). EH with and without an LCD is a therapeutic strategy for improving bone mass in T2DM, which is associated with cardiorespiratory and functional improvements.

## 1. Introduction

The International Diabetes Federation (IDF) estimates that diabetes mellitus will affect 783 million people by 2045 [1] and that over 90% of diabetes mellitus cases are type 2 [2]. Type 2 diabetes mellitus (T2DM) describes a group of metabolic disorders characterized by high blood glucose levels that could promote bone metabolism defects due to osteoblast dysfunction [3], which poses an increased risk of developing several serious, life-threatening health problems resulting in higher medical care costs, reduced quality of life, and increased mortality, mainly from cardiovascular diseases [4,5]. 

An essential consequence of aging in people with T2DM is cardiovascular complications [5] and diminished levels of functional capacity (a set of balance, gait, and coordination elements), especially loss of strength and mobility. Impaired strength has been shown to exert adverse effects on bone health [6], and impaired mobility results in a greater risk of falls and bone fragility, which appear to be important causes of non-fatal injuries among people over 65 years of age [7] and with T2DM [8]. Therefore, functional strength and mobility are central elements of health quality, as they predict functional decline, the use of health services, and morbidity and mortality [9]. Furthermore, older adults with T2DM tend to have diminished levels of cardiorespiratory fitness, which could be associated with losses in bone mineral density since peak oxygen (VO_2peak_) consumption is one of the strongest predictors of bone variables [10] due to its involvement in the activation of genes related to osteogenesis, such as erythropoietin (EPO) [11].

The International Foundation for Osteoporosis has provided guidelines for fracture prevention in diabetes [12] since the fracture risk among individuals with T2DM is up to three times greater than that among non-diabetics [13,14]. Although not yet understood, evidence is growing to suggest that T2DM and osteoporosis share pathophysiological mechanisms [15]. Hyperglycemia increases oxidative stress and inflammation in the skeleton, and reduces the expression of endothelial cells, which decreases blood flow, oxygen supply, and bone vasculature. Also, hyperglycemia alters the enzymatic type I collagen cross-linking by glycation and oxidation to non-enzymatic advanced glycation end products (AGEs) [16], promoting reduced extracellular matrix quality and compromising bone health [17]. It has already been demonstrated that good management of blood glucose levels in patients with T2DM contributes to reducing the risk of fracture by decreasing the accumulation of AGEs in bone tissue, responsible for making bones more fragile and less resistant [14].

Progressive and metabolically unfavorable changes in body composition have long been observed with both aging and T2DM, with an accumulation of fat (especially in the waist area), and with the loss of lean mass. A multifactorial approach, including physical exercise and dietary control, leads to success in losing weight and body fat and treating disturbances in glucose metabolism [18], and is thus able to promote T2DM remission, defined by hemoglobin A1c (HbA1c) < 6.5% [19]. A low-carbohydrate diet (LCD) is described as a favorable dietary intervention for managing glycemic control in patients with T2DM [20,21]. In practice, LCDs restrict grains, cereals, and legumes, and other foods that contain high carbohydrate contents, such as dairy, some fruits, and certain vegetables. The energy required is then typically replaced with food higher in fat, such as oleaginous fruits, butter, cream, and oils [22]. However, there is no agreed definition for LCDs. Hence, there is variation across studies [23], with the reported benefits in glycemic control ranging from 21 g daily to 45% of the daily energy intake (225 g for 2000 kcal) [22,24].

Being physically active may reduce the risk of developing T2DM by approximately 50% [25] and improve bone health [26] and functionality [27]. More recently, the therapeutic benefits of exercise in hypoxia (EH) have been suggested for clinical populations, with dose being a relevant factor [28]. In this context, moderate hypoxia, between 2500 m and 3000 m of simulated altitude, most often leads to beneficial effects [29]. It is suggested that specific adaptations induced by hypoxia are mediated mainly by the increase in the hypoxia-induced transcription factor 1α (HIF-1α), which, in addition to offering potentially therapeutic effects for bone health by inducing an osteogenic–angiogenic response [28], promotes improvements in glucose metabolism in patients with T2DM to a greater extent than in normoxic conditions [30]. In older adults, EH has shown to be beneficial to cognitive performance and quality of life [31], functional capacity [32], and bone health [33]. These adaptations raise the question of the usefulness of EH as a therapeutic intervention in T2DM, but the evidence remains scarce.

Thus, the present study aimed to investigate the effects of eight weeks of normobaric EH at 3000 m of simulated altitude combined with an LCD (40% of total energy intake from carbohydrates) on the HbA1c, body composition, and functional and physiological capacity of older adults with T2DM. We hypothesized that a combination of chronic EH and LCD treatment in patients with T2DM is associated with improved glycemic control, body mass index (BMI), body fat and bone mass content (BMC), cardiorespiratory fitness, functional mobility, and strength in T2DM.

## 2. Materials and Methods

### 2.1. Ethical Considerations

This randomized controlled trial (RCT; NCT05094505) strictly followed informed consent, confidentiality, and anonymity protocols. An informed consent form containing guidelines, the project objectives, a description of the procedures, the possible risks, and the possible benefits was provided, which was in agreement with and approved by the Ethics Committee of the Faculty of Nutrition and Food Sciences, University of Porto—reference 45/2021/CEFCNAUP/2021. Also, the Declaration of Helsinki for studies in humans was applied and followed [34]. The participants were also informed that their participation was voluntary and they could withdraw from the study at any time without prejudice. In addition, if requested, the participants could access their data after the trial by contacting the lead researcher.

### 2.2. Study Design

This was an eight-week, controlled, single-blind, three-arm, parallel RCT. As a sample, 42 participants were included in the study, according to the result obtained through the sample calculation. Participants were randomly assigned into three independent groups (*n* = 14, in each): a control group (control diet + exercise in normoxia), an EH group (control diet + EH), and an EH + LCD group. The experimental design consisted of (1) one week of pre-intervention tests, (2) two weeks of a familiarization period, (3) eight weeks of experimental intervention, and (4) one week of post-intervention tests (Figure 1).

### 2.3. Participants

The eligibility criteria for inclusion in the study were as follows: (1) elderly male or female (age > 65 years) patients with a previous T2DM diagnosis, having had this diagnosis for at least one year; (2) HbA1c ranging from 6.5% to 10%; (3) pharmacological regimen stabilized for at least three months; (4) previous participation in supervised exercise programs in the last six months; and (5) the absence of smoking in the last six months. Participants were excluded if (a) they were insulin-dependent; (b) they had uncontrolled microvascular or macrovascular complications related to diabetes; (c) they presented with other uncontrolled metabolic or vascular comorbidities; (d) they were inactive; or (e) they had a physical limitation that prevented them from exercising.

All criteria were verified and confirmed through interviews with individual questionnaires, biochemical analyses, and physical activity records provided by the sports programs where the recruitment was carried out.

### 2.4. Dietary Plan

Each participant was prescribed an individualized dietary plan using the Dietbox^®^ software, version 7.0. The energy content of the dietary plan met 100% of the estimated energy requirement (EER) for each participant. The EER was calculated by multiplying the resting metabolic rate obtained from the Harris–Benedict equation, the most accurate at the individual level for older adults [35], by the physical activity level, assessed using the International Physical Activity Questionnaire (IPAQ, short form, last seven days, elderly, self-administered format). Participants were categorically rated into one of three levels of physical activity—low, moderate, or high [36].

Regarding the energy distribution across the macronutrients, 60% came from carbohydrates, 20% came from protein, and 20% came from fat for the control diet; and 40% of the energy came from carbohydrates, 20% came from protein, and 40% came from fat for the LCD diet. Both diets emphasized low-glycemic-index foods to convey conventional dietary guidelines [37]. We used the following classifications for the diets: (1) very-low-carbohydrate: less than 26% of the energy intake; (2) low-carbohydrate: 26–45% of the energy intake; and (3) high-carbohydrate: ≥ 45% of the energy intake [38]. Thus, we compared a low-carbohydrate diet (EH + LCD group) to a high-carbohydrate diet (EH group). Weekly 24 h recalls were applied to assess the compliance with the dietary plan. Participants met individually in appointments with a nutritionist two times over the eight weeks to stimulate their adherence to the dietary plan.

### 2.5. Exercise Protocol

Exercise sessions, either in normoxia or hypoxia via nitrogen dilution, at 3000 m of simulated altitude, took place in a hypoxic chamber at CMEP—Exercise Medical Center three times per week during an intervention period of eight weeks. The chamber allowed the control of O_2_ (11% to 20.97%), temperature (up to 50 °C), relative humidity (up to 80%), and altitude above sea level (up to 8000 m). The altitude rate can be defined as (1) high altitude: 1500 to 3500 m; (2) very-high altitude: 3500 to 5500 m; or (3) extreme altitude: above 5500 m [39].

Before starting the intervention period, six familiarization sessions were held for two weeks so that the participants could learn the exercise techniques and acclimate to the simulated altitude, with an increment of 500 m at each visit until reaching 3000 m of altitude. The exercise intensity was set at 75% of the heart rate reserve, quantified in the pre-intervention cardiopulmonary test (CPET). Heart rate and oxygen saturation were constantly monitored using a finger pulse oximeter (Globus YM201, Milan, Italy), as well as the Borg Rating of Perceived Exertion (RPE) being recorded after each exercise session. All exercise sessions occurred at the same time of day (±1 h), and visits were separated by at least 48 h of recovery.

The total duration of each exercise session was approximately 60 min, which included a 5 min warm-up with body mobilization and dynamic stretching, followed by 40 min of moderate aerobic exercise, alternated every 9 min on a cycle ergometer (Life Fitness, Chicago, IL, USA) and a treadmill (Life Fitness, Chicago, IL, USA), with a 1 min rest between them. At the end of each session, and alternately between weeks, three strength exercises were performed, including core, inferior, and superior members and using 3 series of 12 to 15 repetitions per exercise, with a 1 min rest between sets, totaling ~15 min of strength exercises.

### 2.6. Measurements

In all groups, the outcomes of glycemic control, body composition, cardiorespiratory, and functional tests were evaluated at baseline and 48 h after the last exercise session (eighth week). A set of clinical outcomes including HbA1c, blood glucose, insulin, BMI, waist–hip ratio, body fat, lean mass, and BMC were collected and evaluated after fasting for 12 h, without any strenuous exercise in the last 24 h and with no alcohol consumption in the previous 72 h. Also, a set of physiological and functional performance outcomes, including VO_2peak_, maximum heart rate, time to exhaustion, peak workload, basal and peak lactate, maximum distance covered in 6 min, and handgrip strength, were collected and evaluated without any strenuous exercise in the last 24 h and with no alcohol consumption in the previous 72 h.

### 2.7. Glycemic Control Analyses

Blood samples were collected after fasting for 12 h, without strenuous exercise in the last 24 h, and with no alcohol consumption in the previous 72 h. Glycemic profiles, namely HbA1c (%) and fasting glucose (mg/dL), were measured, and HOMA-IR estimated insulin resistance.

### 2.8. Body Composition

Height was measured using a stadiometer with 0.1 cm intervals and weight was evaluated with a digital scale with capacities ranging from 0.1 kg to 150 kg (Seca, Hamburg, Germany). Waist and hip circumferences were measured with inelastic tape, between the iliac crest and the last rib, and around the larger perimeter of the buttocks, respectively [40]. The ratio between the waist and hip circumferences was obtained. BMI was calculated and expressed in kg/m^2^. Lean mass, fat mass, android and gynoid fat depots, and BMC were assessed using dual-energy X-ray absorptiometry (DEXA, Hologic HORIZON Wi; APEX System Software Version 5.6.1.3, Hologic Inc., Bedford, MA, USA). Participants were instructed to refrain from performing other procedures with contrast or radiation the day before the measurements. The procedure was conducted in the supine position after 12 h of fasting, with bare feet, light clothes, and no metallic objects.

### 2.9. Cardiopulmonary Exercise Test

The Bruce treadmill exercise test protocol [41] was performed with all participants at baseline and 48 h after the last exercise session (eighth week), using electrocardiography and a face-mask device to evaluate gas exchange adaptations (Cortex Metalyzer 3B^®^, Leipzig, Germany). The test began with walking at 2.7 km/h, with a gradual increase in speed and inclination every 3 min, until the test was completed at the participant’s request or upon medical advice. Exercise tolerance and cardiac and ventilatory adaptations were obtained, and a report with clinical conclusions was made. Our study measured heart rate (bpm), VO_2peak_, time to exhaustion (s), and peak workload (W; km/h). Also, baseline and peak lactate were collected from the fingertip (25 µL, Lactate Pro, Arkay, Inc., Kyoto, Japan).

### 2.10. Functional Tests

This study used the standard procedure for 6MWT of the American Thoracic Society Guidelines [42]. The test occurred before and 48 h after the last exercise session (eighth week) and was held indoors, and participants were instructed to walk as quickly as possible to make the longest distance within a 6 min timeframe. The assessment of muscle strength was made based on handgrip strength. Measurement was performed with the dominant hand using a handheld Jamar^®^ dynamometer (Avantor, Radnor, Pennsylvania), with the participant seated with their elbow flexed at 90°. The participant was instructed to squeeze the handle as hard as possible for six seconds. This measurement was repeated thrice with a recovery period of one minute between measures and was registered in kgf. The mean value was calculated and used for the statistical analysis.

### 2.11. Statistical Analyses

For sample and power calculations, this study was powered based on changes in HbA1c in the RCTs included in the meta-analysis by Zuuren et al. [43]. Statistical analyses were performed using SPSS Statistics software version 28.0, 2021 (IBM Company, Chicago, IL, USA). Normality was assessed using Shapiro–Wilk’s test. Continuous variables that did not follow a normal distribution were transformed using the logarithm function: y = log(x − L) with L < minimum of x, if the skewness was positive; or y = log(H − x) with H > maximum of x, if the skewness was negative. A one-way ANOVA was used to assess differences between the RPE, oxygen saturation, and diets. A two-way repeated-measures ANOVA was used to examine changes in the physiological and functional tests, body composition results, and glycemic profiles over the chronic exercise period (zero vs. eight weeks) and whether the magnitude of the chronic exercise-mediated adaptations differed across time or differed among the groups. A Tukey post hoc test for multiple pairwise comparisons was performed to identify differences between groups when a significant main effect or interaction effect was found. All data were reported as the mean (standard deviation, SD), and statistical significance was assumed at *p* ≤ 0.05.

## 3. Results

### 3.1. Baseline Characteristics

A total of 42 older adults with T2DM completed the interventions and all of the assessments and were subsequently included in the analysis. None of the participants became injured or had adverse responses to the EH or LCD. Table 1 presents the participant variables that were obtained at the beginning of the study. The groups did not differ significantly at this moment.

### 3.2. Dietary Intervention

The EH + LCD group presented statistically lower values for carbohydrate intake (*p* < 0.001) and higher values for the intake of total fats (*p* < 0.001), monounsaturated fats (*p* < 0.001), polyunsaturated fats (*p* < 0.001), and saturated fats (*p* <0.001). There was no difference among the groups regarding energy (*p* = 0.69) and fiber (*p* = 0.49) consumption, as shown in Table 2.

### 3.3. Physiological Measurements during Exercise Sessions

During eight weeks of exercise sessions, the groups that exercised in hypoxia, EH and EH + LCD, presented lower mean values of oxygen saturation when compared to the CTRL group (*p* < 0.001), and the average heart rates were similar among groups (*p* = 0.63).

### 3.4. Subjective Effort Perception

The average values obtained after 24 exercise sessions demonstrated that the groups that trained in hypoxia, EH and EH + LCD, achieved the highest scores for RPE (*p* ≤ 0.001), as shown in Table 3.

### 3.5. Glucose Metabolism

After eight weeks of intervention, HbA1c and blood glucose decreased in all groups (*p* < 0.001; *p* = 0.019, respectively), and no effect was seen on HOMA-IR (Table 4). While HbA1c and blood glucose levels decreased after the interventions, post hoc analysis revealed that these changes did not reach statistical significance between the groups (*p* = 0.090; *p* = 0.977, respectively).

### 3.6. Anthropometry and Body Composition

The anthropometric characteristics and body compositions of the study participants are shown in Table 5. All groups decreased their weight (*p* < 0.001), BMI (*p* < 0.001), waist and hip circumferences (*p* < 0.001; *p* = 0.029), waist–hip ratio (*p* = 0.028), and body fat (*p* < 0.001). Moreover, BMC increased in all groups after the eight weeks compared to baseline values (*p* < 0.001). Significant differences between groups were detected for BMC (*p* < 0.001), with higher values in the groups exercising in hypoxia. There were no significant differences in lean mass after the eight weeks of exercise and diet interventions (*p* = 0.388).

### 3.7. Functional Capacity

As shown in Table 6, the handgrip strength and maximum distance covered increased in all groups (*p* < 0.001), with larger improvements in the 6MWT after exercising in hypoxia (*p* = 0.030).

### 3.8. Cardiorespiratory Fitness

From pre to post intervention, VO_2peak_ improved in all groups (*p* < 0.001), more markedly in the hypoxia groups, regardless of the diet (*p* = 0.019). Also, the time to exhaustion and peak workload improved in all groups after interventions (*p* < 0.001), and the basal heart rate decreased (*p* = 0.020). On the other hand, the maximum heart rate decreased in the CTRL and EH groups (*p* = 0.008) and increased in the EH + LCD group (*p* = 0.023). Both the basal (*p* < 0.001) and lactate peak (*p* = 0.001) decreased from baseline levels without significant differences between the three groups (*p* = 0.183 and *p* = 0.690, respectively), as shown in Table 6.

## 4. Discussion

To our knowledge, this is the first randomized controlled clinical trial focusing on the effects of chronic EH and LCD treatment in patients with T2DM. The study’s main findings were the improvements in VO_2peak_ and BMC after eight weeks of training in hypoxia, regardless of the carbohydrate content of the diet. Despite the improvements observed, these outcomes contradict our hypothesis regarding the greater effects of combined interventions in relation to their isolated application. This study also demonstrated improvements in body fat, waist–hip ratio, glucose parameters, capillary lactate, handgrip strength, and maximum distance covered in 6 min after intervention, with the last one being more evident in the EH group.

Interestingly, positive relationships have already been demonstrated in skeletal systems, with increased cardiorespiratory fitness [44], reduced body fat [45], improved functionality through increased handgrip strength [46], and through glycemic control [47]. All of these data corroborate the results presented in our study, which, in a unified way, can justify the improvements in bone mass obtained after eight weeks of EH.

It is well known that a pro-inflammatory state, common in patients with T2DM due to hyperglycemia [48], is an important determinant of reduced bone quality. On the other hand, an inverse relationship between cardiorespiratory fitness and inflammatory markers was described in adults with metabolic syndrome [44]. Furthermore, it has been shown that reductions in BMI, body fat, and HbA1c may reduce bone resorption activity, while cardiorespiratory fitness may increase bone formation activity in patients with T2DM [47]. It is noteworthy that in our study, BMI, waist circumference, and body fat were reduced after the exercise and diet interventions compared to before in all groups, and they were inversely associated with cardiorespiratory fitness, which is consistent with previous data [49]. Also, greater improvements were found in VO_2peak_ in patients with T2DM who performed aerobic and resistance exercises during six weeks in hypoxia [50], but not in older adults who performed resistance EH during eight weeks [51]. These data could suggest the greater efficiency of the combination of aerobic and resistance exercises relative to their isolated application, as suggested by the World Health Organization [52].

This corroborates previous studies that have indicated that the association between functional capacity based on handgrip strength and BMC is site-specific [53] and systemic [54]. In this context, it was reported that handgrip strength of the dominant hand was an independent factor for BMC and was positively associated with the bone system [53]. Li et al. [55] found that postmenopausal female patients aged >65 years with lower handgrip strength had significantly lower bone mass. It has been suggested that reduced handgrip muscle strength might be a significant factor for bone cortical porosity development in patients with T2DM [8]. Therefore, handgrip strength has been used as an important index of low muscle strength to diagnose sarcopenia associated with low BMC and osteoporosis among middle-aged and older men [56] and female patients with T2DM [8].

In this context, it has already been confirmed that functional performance using 6MWT is correlated with aerobic capacity and muscular fitness [57], partially justifying data from our study. However, there remains no evidence correlating improvements in 6MWT with bone metabolism. Recently, it was demonstrated that intermittent EH for twenty weeks did not influence functional capacity in the elderly, but concomitantly improved inflammatory biomarkers, bone formation, and bone mass, suggesting that it is an important therapeutic strategy for preventing damage to the skeletal system caused by advancing in age [32].

The increase in oxidative capacity induced by exercise and hypoxia, evidenced by lower lactate production, suggests an improvement in both aerobic and anaerobic functional capacity, as assessed with the 6MWT handgrip strength test, respectively. This increase also improves physiological parameters, such as VO_2peak_ and the basal heart rate, with attenuation of fatigue. It is important to highlight that although handgrip strength is primarily an anaerobic test, the systemic and muscular improvements resulting from increased oxidative capacity through aerobic exercise contribute to better overall muscular performance and endurance [58]. Also, increased lactate levels are common in patients with T2DM, due to an imbalance in glucose metabolism and lactate transporters [59]. Thus, a close relationship between lactate metabolism and the manifestation of disturbances in the insulin response appears, which could reduce bone formation [60]. Therefore, in our study, lower lactate levels could suggest better glycemic control [61], as confirmed by the reduced levels of HbA1c, which could be positively associated with bone strength by reducing the deposition of AGEs [62], and consequently reducing oxidative stress and the expression of mediators of inflammation [63]. Unfortunately, our study did not evaluate AGEs, inflammation, or oxidative stress markers.

In agreement, it was demonstrated that acute aerobic EH at a simulated altitude of 3000 m (FiO_2_ = ~14,7%) improved glycemic control in patients with T2DM [64]. However, aerobic and strength EH at 2000 m of simulated altitude (FiO_2_ = ~16.5%) did not seem to be sufficient to obtain improvements after eight weeks, both in the glycemic profile and cardiorespiratory capacity, in this same population [65]. Therefore, EH’s benefits may occur with simulated altitudes over 3000 m [30]. Improvements in both the glycemic profile and VO_2peak_ seem to influence improvements in bone metabolism in patients with T2DM [44]. Many physiological aspects obtained with EH can also play an important role in the skeletal system [66].

EH has been shown to promote physiological changes in bone tissue in older adults, evaluated by the most sensitive and clinically helpful markers of bone resorption, such as serum C-terminal telopeptide of type I collagen (CTX), and of bone formation, such as N-terminal propeptide of type I procollagen (PINP), after 24 weeks of resistance EH [67]. In healthy males, high-intensity resistance EH over eight weeks was not found to cause significant improvements in BMC, despite increasing the exercise in normoxia by 1.21% [68], suggesting, at least in part, that the magnitude of the effect of EH on BMC may be associated with increasing age. Also, both the type and duration of hypoxic exercise influence bone remodeling. Performing exercises in hypoxia for a short period appears to promote greater bone formation and, on the other hand, less bone resorption. In addition, the combination of strength and aerobic exercises demonstrated greater efficiency in the production of osteoblasts and bone formation compared to exclusively aerobic exercises [69].

This is the first study demonstrating the benefits of combining aerobic and resistance EH during eight weeks on BMC and VO_2peak_ in patients with T2DM. Hypoxia-regulated transcription activates genes and pathways that reduce oxygen consumption and cellular dependence on oxygen [70,71]. At the same time, HIF-1α stabilization could activate different genes involved in bone remodeling, such as vascular endothelial growth factor (VEGF), osteoprotegerin (OPG), and EPO. VEGF plays a concomitant and essential role in angiogenic and osteogenic responses. In addition, EPO has been shown to stimulate bone formation and repair, as well as being capable of promoting a physiological increase in cardiorespiratory capacity [11].

It is important to mention that although long-term continuous hypoxia has adverse effects on bone metabolism, favoring bone resorption [72], short periods of exposure to hypoxia, followed by periods of normoxia, can modulate the differentiation of mesenchymal stromal cells and improve bone health in older adults [73]. This occurs due to an increase in bone formation from osteoblast activity, which is promoted by reducing the ratio of receptor activator to nuclear factor kappa B ligand (RANK-L) and OPG. This relationship is an important determinant of the regulation of bone metabolism, and its increase can impair the skeleton by providing greater osteoclast activity and bone resorption, in addition to inhibiting osteoblastic activity and bone formation [73]. In this way, intermittent exposure to hypoxia could ensure benefits to the skeletal system.

Our study demonstrated that, among T2DM patients, moderate, short-term, and intermittent EH increases BMC and VO_2peak_ to a greater extent than exercise in normoxia. Although still scarce, the existing evidence demonstrates the superior benefits of EH compared to exercise in normoxia on the skeletal system, opening a potential scientific gap to be explored not only in the elderly and patients with T2DM but also in populations with osteoporosis and other bone disorders.

Our study has several strengths. Meetings were held to evaluate food consumption and clarify any doubts for all participants to encourage correct adherence to the dietary plan. By ensuring chauffeured transportation for the participants to exercise sessions at the clinic, the study had a 100% adherence rate, with no dropouts throughout the testing and intervention phases, presenting 30% above the general cutoff point used for sufficient adherence in older adults [74]. Furthermore, the research team maintained constant contact with the participants, motivating them to complete their participation regularly during the eight weeks of intervention.

On the other hand, some limitations should also be mentioned. Due to logistical and budgetary reasons, the study had a short duration of eight weeks. The authors believe twelve weeks of intervention could reveal differences in HbA1c between the groups because of its 120-day half-life [75]. Considering the reasons mentioned above, HIF-1α, VEGF, OPG, EPO, and AGE levels were not determined, presenting an essential line of research to explore in the future.

## 5. Conclusions

The combination of aerobic and strength exercises performed in hypoxia for eight weeks significantly increases bone mass, cardiorespiratory capacity, and functional aerobic performance in elderly patients with T2DM, regardless of their carbohydrate content of their diet. Future research is needed to provide a deeper understanding of the physiological mechanisms involved in skeletal system responses.

## Figures and Tables

**Figure 1 nutrients-16-01624-f001:**
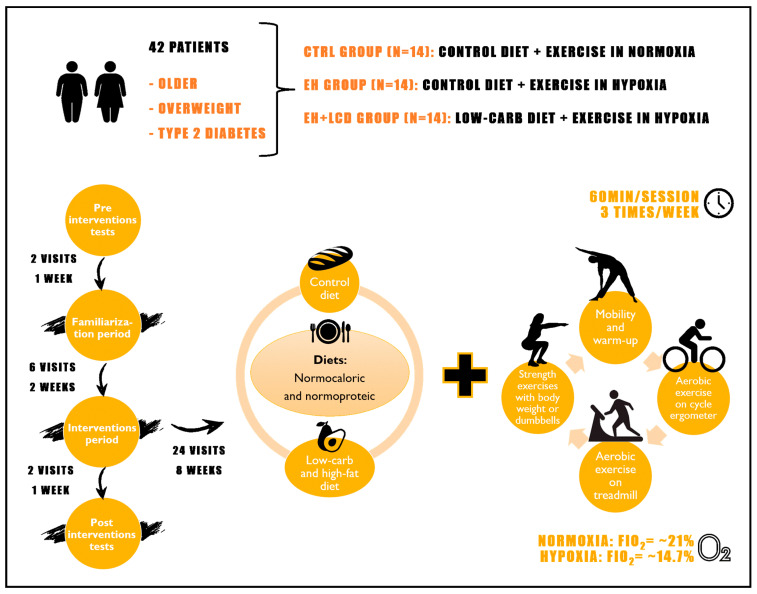
Illustrative scheme of the randomized clinical trial design.

**Table 1 nutrients-16-01624-t001:** Baseline characteristics of the T2DM study participants.

Variable (*n* = 42)	Value
Gender (male–female)	17:25
Age (years)	72.2 (4.0)
Body mass index (kg/m^2^)	29.0 (3.8)
Body fat (%)	39.0 (6.5)
Waist circumference (cm)	96.7 (8.4)
Hip circumference (cm)	103.8 (14.1)
Waist–hip ratio (cm)	0.9 (0.1)
Hemoglobin A1c (%)	7.1 (1.7)
Fasting glucose (mg/dL)	114.3 (24.3)
HOMA-IR	2.4 (1.6)

In total, 42 subjects participated in this study. These anthropometric and glycemic data were taken at the beginning of the study. Data are presented as the mean (standard deviation, SD).

**Table 2 nutrients-16-01624-t002:** Nutrient intakes for the control diet groups (CTRL and EH) and carbohydrate-restricted diet group (EH + LCD), during eight weeks of intervention, in T2DM patients.

Nutrients	CTRL Group	EH Group	EH + LCD Group	*p*-Value
Energy (kcal)	1603.7 (107.1)	1666.8 (246.9)	1619.2 (155.4)	0.69
Carbohydrates (%)	56.7 (3.2)	56.6 (2.7)	39.5 (0.6) #	<0.001
Protein (%)	19.4 (2.3)	20.8 (1.0)	20.1 (0.7)	0.15
Total lipids (%)	23.7 (2.6)	22.5 (2.8)	40.4 (0.4) #	<0.001
Monounsaturated fat (g)	13.3 (5.1)	12.9 (5.8)	31.5 (9.1) #	<0.001
Polyunsaturated fat (g)	7.4 (1.5)	8.0 (1.6)	13.9 (4.8) #	<0.001
Saturated fat (g)	9.8 (3.4)	7.6 (1.5)	13.2 (2.) #	<0.001
Fiber (g)	20.7 (3.9)	24.6 (11.9)	22.3 (4.3)	0.49

Data are presented as the mean (standard deviation, SD), after analysis with a one-way ANOVA for comparison among groups. The symbol “#” indicates when *p* < 0.05 in relation to the EH + LCD group.

**Table 3 nutrients-16-01624-t003:** Mean values of oxygen saturation, heart rate, and subjective perception of exertion according to the Borg Rating, obtained in each session and during an eight-week period of exercise, in patients with T2DM.

Variable	CTRL Group	EH Group	EH + LCD Group	*p*-Value
Oxygen saturation (%)	94.8 (2.8)	87.1 (4.6) #	87.2 (4.4) #	<0.001
Average heart hate (bpm)	115.5 (7.7)	120.7 (9.5)	118.5 (11.8)	0.63
Borg Rating of Perceived Exertion (points)	12.6 (0.5)	14.8 (1.6) #	15.0 (1.3) #	<0.001

Data are presented as the mean (standard deviation), after analysis with a one-way ANOVA for comparison among groups. The symbol “#” indicates when *p* < 0.05 in relation to the CTRL group.

**Table 4 nutrients-16-01624-t004:** Glycemic responses pre and post eight-week intervention in patients with T2DM.

Variables	CTRL Group			EH Group			EH + LCD Group			*p*-Value	
	Pre	Post	Δ	Pre	Post	Δ	Pre	Post	Δ	Moments	Groups
HbA1c (%)	6.9 (0.8)	6.7 (0.7)	0.4 (0.4)	7.1 (0.7)	6.7 (0.6)	0.7 (0.5)	6.8 (0.5)	6.4 (0.6)	1.2 (1.3)	<0.001 *	0.090
Glucose (mg/dL)	118.7 (27.8)	110.8 (22.3)	2.8 (8.9)	117.9 (22.3)	111.7 (18.8)	2.3 (6.2)	108.2 (19.7)	103.1 (20.5)	2.4 (4.3)	0.019 *	0.977
HOMA-IR	2.2 (1.4)	2.2 (1.0)	0.1 (0.6)	3.4 (2.0)	2.7 (1.7)	0.1 (0.2)	1.5 (0.5)	1.7 (0.8)	0.1 (0.3)	0.923	0.310

Data are presented as the mean (standard deviation). A post hoc Tukey test was used to assess differences between groups. *p*-values represent a two-way repeated-measures ANOVA; “moments” compare the overall means for the pre and post evaluations; and “groups” is the interaction term comparing the time variations among the groups. * = significant differences between results pre and post eight-week intervention. Δ = Changes from baseline to the eighth week.

**Table 5 nutrients-16-01624-t005:** Anthropometric parameters and body compositions pre and post eight-week intervention in patients with T2DM.

Variables	CTRL Group			EH Group			EH + LCD Group			*p*-Value	
	Pre	Post	Δ	Pre	Post	Δ	Pre	Post	Δ	Moments	Groups
Weight (kg)	74.9 (8.1)	73.7 (7.0)	0.4 (0.6)	72.6 (12.4)	72.0 (12.5)	0.2 (0.3)	73.5 (9.6)	72.0 (9.6)	0.4 (0.4)	<0.001 *	0.153
Body mass index (kg/m^2^)	29.4 (4.1)	28.8 (3.6)	0.5 (0.6)	28.3 (4.0)	28.0 (3.9)	0.2 (0.2)	29.3 (3.4)	28.5 (3.3)	0.8 (0.8)	<0.001 *	0.087
Waist circumference (cm)	96.6 (5.6)	95.0 (5.7)	2.6 (2.5)	97.0 (10.6)	95.0 (10.6)	2.0 (2.0)	95.6 (8.6)	93.6 (8.4)	2.0 (2.7)	<0.001 *	0.752
Hip circumference (cm)	102.7 (7.4)	102.0 (7.2)	1.5 (2.9)	102.9 (10.1)	101.9 (9.1)	1.1 (5.6)	103.5 (6.5)	102.6 (6.1)	1.6 (3.0)	0.029 *	0.951
Waist–hip ratio	0.9 (0.1)	0.9 (0.1)	0.1 (0.1)	0.9 (0.8)	0.9 (0.1)	0.1 (0.1)	0.9 (0.1)	0.9 (0.1)	0.1 (0.1)	0.028 *	0.986
Body fat (%)	38.4 (6.9)	37.5 (6.7)	0.8 (1.1)	38.8 (7.3)	38.0 (7.7)	0.8 (1.4)	39.8 (5.6)	38.5 (6.2)	1.3 (1.2)	<0.001 *	0.589
Body fat (kg)	27.6 (7.7)	26.2 (7.0)	0.5 (0.4)	28.2 (8.3)	27.8 (8.1)	0.1 (0.4)	29.1 (6.2)	27.8 (6.5)	0.5 (0.4)	<0.001 *	0.079
Lean mass (kg)	43.3 (4.2)	43.3 (4.0)	0.1 (0.9)	41.9 (7.8)	42.3 (8.3)	0.3 (1.4)	41.1 (6.2)	41.1 (6.3)	0.1 (0.6)	0.388	0.624
Bone mineral content (kg)	1.8 (0.3)	1.8 (0.3)	0.1 (0.1)	1.9 (0.3)	2.1 (0.4)	0.1 (0.1)	2.0 (0.3)	2.1 (0.3)	0.1 (0.1)	<0.001 *	<0.001 #

Data are presented as the mean (standard deviation). The post hoc Tukey test was used to assess differences between groups. *p*-values represent a two-way repeated-measures ANOVA; “moments” compare the overall mean for the pre and post evaluations; and “groups” is the interaction term comparing the time variations among the groups. * = Significant differences between results pre and post eight-week intervention. #: Bone mineral content: EH and EH + LCD groups increased more than the CTRL group. Δ = Changes from baseline to eighth week.

**Table 6 nutrients-16-01624-t006:** Functional and physiological capacity pre- and post-eight-week interventions in patients with T2DM.

Variables	CTRL Group			EH Group			EH + LCD Group			*p*-Value	
	Pre	Post	Δ	Pre	Post	Δ	Pre	Post	Δ	Moments	Groups
**Functional capacity**											
Maximum distance covered (m)	414.6 (62.0)	441.1 (70.0)	26.5 (49.1)	430.9 (98.8)	505.0 (87.7)	74.0 (46.0)	456.1 (70.9)	504.9 (101.0)	48.7 (58.9)	<0.001 *	0.030 #
Handgrip of dominant hand (kg)	23.5 (8.8)	24.7 (7.7)	1.2 (2.5)	23.5 (8.8)	26.9 (7.9)	3.4 (3.2)	23.0 (7.2)	25.5 (8.0)	2.4 (3.4)	<0.001 *	0.234
**Physiological capacity**											
Peak oxygen uptake (mL O_2_/min/kg)	21.4 (3.9)	24.5 (4.5)	3.8 (3.4)	21.8 (3.6)	29.2 (3.9)	10.0 (4.9)	19.6 (6.5)	27.7 (9.7)	7.4 (5.0)	<0.001 *	0.019 #
Basal heart rate (bpm)	77.6 (11.0)	73.7 (12.6)	6.1 (6.9)	75.2 (12.2)	70.8 (11.2)	3.8 (3.1)	73.5 (10.8)	72.8 (6.9)	0.4 (8.9)	0.020 *	0.134
Maximum heart rate (bpm)	127.8 (11.7)	121.8 (13.9)	8.0 (12.8)	149.1 (18.3)	132.5 (24.0)	13.5 (14.2)	129.2 (22.2)	132.0 (23.2)	2.6 (10.4)	0.008 *	0.023 #
Time to exhaustion (s)	344.6 (114.6)	489.0 (139.7)	155.0 (141.5)	536.2 (83.6)	629.5 (144.7)	124.5 (112.5)	460.4 (113.8)	684.1 (179.3)	217.5 (139.1)	<0.001 *	0.197
Peak workload (Watt)	178.4 (31.6)	248.1 (72.4)	76.5 (70.4)	232.8 (47.9)	283.7 (47.4)	54.1 (34.3)	230.2 (50.9)	249.4 (88.8)	32.7 (68.5)	<0.001 *	0.131
Peak workload (km/h)	4.6 (0.7)	5.8 (0.9)	1.3 (1.1)	5.7 (1.0)	6.6 (1.3)	0.9 (0.6)	5.6 (0.8)	6.5 (1.3)	1.1 (1.1)	<0.001 *	0.706
Basal lactate (mmol/L)	6.6 (1.9)	5.2 (1.8)	1.3 (1.3)	6.2 (2.5)	4.4 (2.6)	2.3 (1.5)	6.4 (2.2)	5.2 (3.2)	1.1 (2.3)	<0.001 *	0.183
Peak lactate (mmol/L)	9.3 (5.0)	6.1 (2.3)	3.2 (4.9)	8.6 (5.8)	6.8 (3.5)	2.2 (2.9)	7.9 (3.7)	6.5 (4.4)	1.4 (3.0)	0.001 *	0.690

Data are presented as the mean (standard deviation). The post hoc Tukey test was used to assess differences between groups. *p*-values represent a two-way repeated-measures ANOVA; “moments” compare the overall mean for the pre- and post-evaluations; and “groups” is the interaction term comparing the time variations among the groups. * = significant differences between results pre- and post-eight-week intervention. #: Maximum distance covered (m): the EH group increased more than the EH + LCD and CTRL groups; peak oxygen uptake (mL O_2_/min/kg): the EH and EH + LCD groups increased more than the CTRL group; maximum heart rate (bpm): the EH + LCD group increased more than the CTRL and EH groups. Δ = Changes from baseline to the eighth week.

## Data Availability

The data presented in this study are available on request from the corresponding author due ethical reasons.

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
