# Peer review of "Eight Weeks of Intermittent Exercise in Hypoxia, with or without a Low-Carbohydrate Diet, Improves Bone Mass and Functional and Physiological Capacity in Older Adults with Type 2 Diabetes"

_nutrients, 2024, doi:10.3390/nu16111624_

Round 1
Reviewer 1 Report
Comments and Suggestions for Authors
This study mainly examines the effects of exercise in hypoxia combined with a low carbohydrate diet on body composition and functional and physiologic capacity in T2DM patients. While interesting, I have some important concerns mainly regarding study design and/or the rationale for important experimental conditions.
-Line 86-87. Please rewrite this sentence in a more logical way. The recommended diet, low in CHO and saturated fat and rich in fiber, can be obtained mainly eating whole grains, fruit and vegetables.
-Lines 88-90. Which is the relevance of this comment in the context of the study? After reading this sentence, one could guess that this question would be ascertained in the study. However, and taking into account my next comment, authors decided to provide 40% of CHO.
-Line105. Why 40% of energy from CHO was considered? Nowadays, in the real world, can this percentage define a low CHO diet?
-It would be good to see a figure reporting the study design.
-Lines 130-137. Please, explain inclusion and exclusion criteria related to previous exercise level (except for the last one).
-Line 136. I could guess that, actually, authors would like to indicate “inactive” rather than sedentary.
-Lines 130-137. It is surprising that no criteria focused on age were considered.
-Lines 130-137. Furthermore, authors should explain how they ascertained participants fulfilled all these criteria.
-Lines 235-238. Please, move this content in the participants’ or the study design sections.
-Authors should clearly report the participants’ recruitment procedures.
-Exercise protocol. How were effects of low oxygen availability considered in terms of exercise intensity? An intensity of 75% in normoxia is not the same as in hypoxic conditions. How the authors consider this issue? Additionally, should exercise in hypoxia could suppose a high exercise level, maybe this should be considered in terms of diet, e.g adjusting energy supply to allow proper comparisons.
-Why was an intermittent exercise protocol applied?
-Table 1 and elsewhere. For most values, an only decimal place is much more adequate. I’m sure measure were not performed at the 0.1 mm or the 0.01-year precision levels. Furthermore, please double-check whether the waist to hip ratio is expressed in cm.
-Lin 341. I’m not sure at all whether an improvement in VO2max after training in hypoxia could be considered as a novelty and a main result of the study. Maybe this could be applied to patients with T2DM.
-Lines 385-387 and elsewhere. Please revise this and similar sentences. How an increase in oxidative capacity (aerobic exercise) could improve handgrip strength (completely anaerobic test)?
-Conclusion section. In my opinion this section (lines 458-463) should be rewritten as in the present form supposes a repetition of results.
Reviewer 2 Report
Comments and Suggestions for Authors
This work studied the benefits of combining aerobic and resistance EH on BMC and VO2peak in patients with T2DM. This might be the first such kind of work in elder patients. Although the number of patient are not that high and the duration of test is relatively short, the information collected from this study is a good start on this facet. The writing, data analysis, discussion and conclusion are as accurate as possible and within a reasonable range. Hopefully there will be more patients could be involved and more information like inflammation (very possible to be equivocal) and related biomarkers and biofactors be observed in the following works.
There're several suggestions about the writing.
There are '3.000m' or '3000m' in the manuscript, might '3,000m' be better.
Since the term 'mesenchymal stromal cells' has long been suggested instead of 'mesenchymal stem cells', suggest to use the former for this kind of cells.
Comments on the Quality of English Language
Good.
Round 2
Reviewer 1 Report
Comments and Suggestions for Authors
In my opinion, the manuscript has been largely improved. I would like to thank the authors for their efford and grateful replies.